# The Physicochemical Characteristics and Heavy Metal Retention Capability of Black Liquor Lignin-Based Biochars

**DOI:** 10.3390/molecules28237694

**Published:** 2023-11-21

**Authors:** Zhanghong Wang, Jiale Li

**Affiliations:** 1College of Eco-Environmental Engineering, Guizhou Minzu University, Guiyang 550025, China; 15885794651@163.com; 2Research Center of Solid Waste Pollution Control and Recycling, Guizhou Minzu University, Guiyang 550025, China; 3Key Laboratory of Energy Thermal Conversion and Control of Ministry of Education, Southeast University, Nanjing 210096, China

**Keywords:** lignin, biochar, pyrolysis temperature, Cd(II), adsorption

## Abstract

Due to its high carbon content, lignin, particularly for lignin-containing solid waste, is considered an excellent raw material for the preparation of carbon materials like biochar. To produce high-quality lignin-based biochar (LGBCs), lignin extracted from black liquor was employed to prepare biochar at various pyrolysis temperatures (300~600 °C). The physicochemical properties of LGBCs were assessed using scanning electron microscopy, N_2_ adsorption/desorption, Fourier transform infrared spectroscopy, Raman spectroscopy, and X-ray diffraction. Furthermore, the adsorption capability and potential mechanism of LGBCs in removing Cd(II) were investigated as well. The results indicate that LGBCs produced at higher pyrolysis temperatures exhibit rougher surfaces and more developed pore structures, which facilitate the exposure of numerous active adsorption sites. The adsorption of Cd(II) by LGBCs generally follows the order of LG-300C < LG-400C < LG-500C < LG-600C. According to the Langmuir adsorption isotherm model, the theoretical maximum adsorption capacity of LG-600C for Cd(II) is calculated to be 18.54 mg/g. Adsorption mechanism analysis reveals that the complexation interaction, dependent on the surface functional groups, plays a crucial role in the adsorption of Cd(II) by LGBCs prepared at higher pyrolysis temperatures. This study demonstrates that, by controlling the pyrolysis temperature during biochar preparation, high-quality lignin-based biochar can be readily obtained.

## 1. Introduction

Cellulose and lignin are the two most abundant natural polymers on Earth, which form the main components of biomass, along with hemicellulose. However, cellulose is more extensively studied and applied due to its simple structure, easy degradability, and controllable degradation. On the other hand, lignin has a complex structure consisting of three types of phenylpropane units (guaiacyl, syringyl, and p-hydroxyphenyl) connected by ether bonds (C-O-C) and carbon–carbon bonds (C-C) [1]. It also has diverse linkages (β-O-4, 5-5, α-O-4, β-5, β-β, 4-O-5, and β-1), making it challenging to utilize [2]. As a result, cellulose can be fully utilized most of the time, while lignin is primarily generated as a byproduct in industries such as papermaking and bio-ethanol production [3]. Developing methods for the resource utilization of lignin would greatly contribute to addressing the current challenges in the cellulose industry’s development and provide guidance for lignin’s resource utilization.

Lignin is a valuable source of carbon and is widely used as a raw material for the preparation of carbon materials such as biochar [4,5]. Biochar, on the other hand, possesses developed pores, abundant surface functional groups, strong environmental stability, and unique optical and electrical properties, making it highly versatile in various fields such as catalysis, energy storage, sensing, and environmental remediation [6,7]. The utilization of lignin for biochar production has gained some attention [8,9]. For example, Zheng et al. reported that a nitrogen-doped porous biochar from industrial alkali lignin was successfully prepared and employed as an excellent electrode material for a supercapacitor [10]. Wang et al. developed a biochar prepared from lignin extracted from poplar wood with superior bond breaking ability as a hydrogenolysis catalyst [11]. It is worth mentioning that preparation conditions, especially pyrolysis temperature, determine the physicochemical properties of biochar, and thus have an important impact on the application performance of biochar [12]. Consequently, it is crucial to understand the influence of pyrolysis temperature on the physicochemical properties of biochar to obtain high-quality lignin-based biochar. However, research in this area remains limited and lacks comprehensiveness.

On the other hand, cadmium (Cd) is one of the important pollutants in water bodies. Long-term exposure or excessive intake will pose serious harm to human health, bringing about diseases like “itai-itai disease”, anemia, osteoporosis, and hypertension [13]. Adsorption methods based on low-cost adsorbents are considered important means for treating Cd(II)-contaminated water bodies and Cd(II)-containing wastewater [14]. Biochar, as a green, low-cost, and high-efficient adsorbent, has been widely used in the treatment of Cd(II)-related contaminations [15]. However, research on lignin-based biochar as an adsorbent are still limited and the adsorption process is not yet clear.

Herein, lignin extracted from black liquor was used as raw material for the preparation of biochars (LGBCs) under different pyrolysis temperatures (300~600 °C) in this study, which were then used for the treatment of Cd(II)-containing wastewater. The physicochemical properties of LGBCs were characterized using scanning electron microscopy (SEM), N_2_ adsorption/desorption, Fourier transform infrared spectroscopy (FT-IR), Raman spectroscopy, and X-ray diffraction (XRD). The adsorption capability of the LGBCs was evaluated by examining the influence of initial Cd(II) concentration, contact time, ambient temperature, and solution pH. Additionally, the potential adsorption mechanism of Cd(II) by the LGBCs was investigated according to the physicochemical properties of LGBCs and model fitting for the adsorption process.

## 2. Results

### 2.1. Physicochemical Properties of LGBCs

#### 2.1.1. Morphological and Textural Properties

The SEM results were employed to reveal the morphological characteristics of LGBCs, which are shown in Figure 1. The biochar prepared under 300 °C (LG-300C) shows a relatively smooth surface and few pore structures can be observed. According to the thermogravimetric analysis shown in Appendix A, LG is highly recalcitrant, possessing a wide pyrolysis range of 250~650 °C [16]. The smooth surface can be attributed to the softening and melting of LG at the initial pyrolysis stage at around 300 °C [17]. With the continuous increasing of pyrolysis temperature (>400 °C), the surface of the obtained biochars (LG-400C, LG-500C, and LG-600C) becomes rough and hierarchical pore structures gradually emerge, which benefits from the decomposition of LG, involving a series of complicated depolymerization and polycondensation reactions. It is reported that the biochar that possesses a rough surface can be used as an excellent adsorbent to remove heavy metal ions from wastewater via surface deposition [18]. And, the developed pore structure of biochar causes a large number of surface active adsorption sites to be exposed, which greatly contributes to the superior adsorption capability of the biochar [19].

To quantify the pore structure of the biochars, the N_2_ adsorption/desorption characteristics of LGBCs were investigated and the results are presented in Table 1. LG-300C exhibits a lower specific surface area (3.14 m^2^/g) but a larger average pore size (64.73 nm). By comparison, biochar with a higher specific surface area and a smaller average pore size is obtained as the pyrolysis temperature increases. LG-600C shows the largest specific surface area and the smallest average pore size, which are 89.76 m^2^/g and 5.36 nm, respectively. The trend of the increase in the specific surface area with the pyrolysis temperature coincides with the SEM results. Similar results can be found in previous studies [20,21]. It is worth noting that, although a similar trend is exhibited in these studies (including the present work), the values of the specific surface area of the corresponding lignin-based biochars are heavily different. For example, the specific surface area of the lignin-based biochar prepared at 500 °C in the present work (LG-500C) is 67.85 m^2^/g, which is much higher than the biochar prepared under the same pyrolysis temperature using lignin reported by Wu et al. (1.32 m^2^/g) and Cha et al. (0.42 m^2^/g) [8,22]. On the other hand, Li et al. reports that the biochar prepared at 600 °C using lignin shows a specific surface area of 259 m^2^/g, which is almost 3 times that of LG-600C in our work [20]. The differences in the specific surface area of the biochars prepared under the same pyrolysis temperature may be related to the specific preparation conditions (such as holding time, atmosphere, heating rate, and reactor) and the characteristics of the raw materials (such as the resource of lignin, surface chemistry property, and composition) [8,19,23].

#### 2.1.2. Surface Functional Groups

As mentioned above, the rough surface and porous structure of LGBCs are conducive to exposing a large amount of surface active adsorption sites to efficiently remove contaminants such as heavy metal ions. It is reported that the surface active adsorption sites of adsorbent mainly originate from its surface functional groups and minerals (ash content) [8]. Herein, FT-IR spectroscopy was employed to analyze the surface functional groups of LGBCs (see Figure 2). It can be observed that LGBCs contain abundant surface functional groups involving -OH (stretching vibration at 3450~3380 cm^−1^), C-H (stretching or bending vibration at 2930, 2840, 1460, 1420, and 750~500 cm^−1^), C=O (stretching vibration at 1640–1610 cm^−1^), C=C ( stretching vibration at 1580, 1560, 1540, and 1500 cm^−1^), aromatic C=C (stretching vibration at 1520 cm^−1^), C-O, C-O-C, and Si-O-Si (stretching vibration at 1520, 1460, and 1120~1020 cm^−1^), and S=O, S-C, and metal-O (stretching vibration in the range of 750–500 cm^−1^) [8,21,22,24,25,26,27,28,29,30,31]. It is worth noting that the vibrations of O/S-containing surface functional groups (C=O, C-O, C-O-C, Si-O-Si, S=O, and S-C) and C-H in LGBCs dramatically weaken with the increasing pyrolysis temperature, which is mainly related to the deoxygenation, dehydration, dehydrogenation, and depolymerization reactions of lignin under higher pyrolysis temperatures, whereas the increasing pyrolysis temperature generally results in the enhancement of aromatic C=C of LGBCs. Similar phenomenon can be readily found in previous studies [8,25]. The stronger aromatic C=C peak observed in the higher pyrolysis temperature-derived LGBCs (such as LG-600C) reveals more aromatic structure and fused rings with abundant conjugated π-electrons. Accordingly, both O/S-containing surface functional groups and aromatic C=C can provide lone-pair electrons and serve as effectively active adsorption sites for heavy metal ion removal via the formation of complexes [8].

The variation of surface functional groups can be further evidenced through elemental analysis, Raman analysis, and XRD patterns. As shown in Table 1, the C content of LGBCs increases with the increasing pyrolysis temperature, while other elements including H, O, and S show the opposite trend, i.e., being reduced with the enhancement of pyrolysis temperature. Among them, the S content mainly came from the paper pulping process due to the addition of S-containing salts such as sodium benzenesulfonate and sodium sulfate. Meanwhile, the use of sulfuric acid to extract LG was also an important source of sulfur in LGBCs via the lignin extraction process (which used sulfuric acid as a precipitant). The higher C content of LGBCs in higher pyrolysis temperatures is consistent with their more abundant aromatic structures (aromatic C=C), and the decrease in H, O, and S is attributed to the decomposition of surface functional groups. Moreover, the lower H/C value of LGBCs derived from higher pyrolysis temperatures reveals a higher aromaticity and stability. Thermogravimetric analysis was carried out to further analyze the stability of LGBCs (see Figure 3), which shows that the weight loss of LGBCs decreases from 35.99% for LG-300C to 22.39%, 16.43%, and 7.33% for LG-400C, LG-500C, and LG-600C, respectively. These results reveal that the biochar prepared under the higher pyrolysis temperature possesses a lower weight loss, showing higher stability. This is in agreement with the results from elemental analysis.

According to Raman analysis (see Figure 4), the D band centered at around 1360 cm^−1^ and the G band centered around 1600 cm^−1^ can be evidently observed in LGBCs (except for LG-300C), which are assigned to disordered carbon structures with abundant structure defects and ordered graphitic carbon structures, respectively [32,33]. The ratio of D peak area and G peak area (D_Aera_/G_Aera_), usually employed to reveal the defect degree of carbon materials, is calculated to be 1.56, 1.41, 1.26, and 1.18 for LG-300C, LG-400C, LG-500C, and LG-600C, respectively [32]. In other words, the D_Aera_/G_Aera_ values of LGBCs decrease with the increasing pyrolysis temperature, indicating that structural defects reduce with the increase in pyrolysis temperature. Combined with the results of FT-IR, the decrease in structural defects is mainly caused by the decomposition of O-containing surface functional groups at higher pyrolysis temperatures [30].

Similarly, the XRD patterns in Figure 5 show that the broad peak corresponding to amorphous carbon can be observed at 2θ around 20.6° in LG-300C. However, it shifts to 22.7° as the pyrolysis temperature increases, revealing the formation of turbostratic stacking carbon (002). Meanwhile, the peak at 2θ around 43.2° assigned to graphitic stacking carbon (100) enhances with the increase in pyrolysis temperature. The amorphous carbon and graphitic stacking carbon correspond to abundant hetero-atom-containing surface functional groups and aromatic C=C, respectively, demonstrating that XRD results are in agreement with those of FT-IR.

#### 2.1.3. Minerals

In addition to carbon-related information, the XRD patterns in Figure 5 reveal the presence of minerals as well. It can be observed that the characteristic peaks located at 31.5°, 45.3°, 56.2°, and 75.0°are assigned to the minerals regarding Na_2_CO_3_, Na_2_SiO_3_, SiO_2_, and NaCl. These results are in line with the study reported by Wu et al., where industrial lignin-based biochar was prepared and investigated [8]. It is well established that Na-related minerals (especially for Na_2_CO_3_ and NaCl) in biochar can serve as highly efficient active adsorption sites for the immobilization of heavy-metal-ion-related contaminants [34]. To quantitatively analyze the content of mineral-related active adsorption sites, the proximate analysis and mineral component analysis of LGBCs were performed and the results are shown in Table 2. It can be seen that LGBCs contain relatively high levels of minerals, which gradually increase with the increasing pyrolysis temperature (3.42~6.22%). The mineral component analysis reveals that LGBCs are composed of a large amount of alkali and alkaline earth metals, especially for Na, and the metal content generally increases with the increase in pyrolysis temperature. In other words, the biochar prepared under a higher temperature possesses more active adsorption sites, which may endow its more superior adsorption capability towards the target contaminants.

### 2.2. Adsorption Behavior of LGBCs towards Cd(II)

The effects of operating conditions regarding the initial concentration of Cd(II), contact time, ambient temperature, and solution pH on the adsorption behavior of LGBCs towards Cd(II) were investigated and the corresponding results are shown in Figure 6. It can be seen that all LGBCs possess relatively superior adsorption capability towards Cd(II), which is generally subjected to an order of LG-300C < LG-400C < LG-500C < LG-600C. In other words, the increase in pyrolysis temperature is beneficial to improving the adsorption capability of LGBCs. Similar findings can be observed in the study by Wu et al., where industrial lignin-based biochar was prepared under different pyrolysis temperatures (200~500 °C) via microwave heating and used for Cd(II) adsorption [8]. In addition to lignin-based biochar, biochars produced from other biomass raw materials, such as rice straw, Napier grass, and earthworm manure, show similar results on the effect of pyrolysis temperature as well [35,36,37].

For the effect of the initial concentration of Cd(II) (Figure 6a), the adsorption capacity of LGBCs gradually increases with the increase in the Cd(II) initial concentration, which is mainly beneficial from the higher driving force under the higher Cd(II) initial concentration [35]. According to Figure 6b, the adsorption of Cd(II) is rapidly proceeded within the initial 30 min for LG-300C and LG-400C and 1 h for LG-500C and LG-600C, accounting for 84.64%, 82.73%, 82.90%, and 79.40% of their total adsorption capacity, respectively. This reveals that the initial stage is crucial for Cd(II) adsorption of LGBCs. Afterwards, the adsorption slows down and a rough equilibrium can be reached in 1h for LG-300C and LG-400C and 2 h for LG-500C and LG-600C. The higher adsorption rate at the initial stage for LGBCs mainly corresponds to their abundant accessible active adsorption sites, whereas the consumption of active adsorption sites results in the decrease in adsorption rate [14]. Moreover, the relatively fast equilibrium obtained (1~2 h) in comparison to bean-worm skin biochar (20 h) and platanus fallen leaves biochar (24 h) suggests that LGBCs can be developed as promising adsorbents for Cd(II) removal [38,39]. The results for the effect of ambient temperature on the Cd(II) adsorption of LGBCs are presented in Figure 6c, which demonstrates that the increasing ambient temperature leads to the enhancement of Cd(II) adsorption capacity, indicating that the adsorption of Cd(II) on LGBCs subjects it to an endothermic process, and higher ambient temperature is in favor of the adsorption. Tan et al. reported a similar result as rice straw biochar was employed to remove Cd(II)-containing wastewater [40]. The adsorption of Cd(II) on LGBCs shows a steady decreasing trend with the decrease in solution pH when the solution pH is lower than the natural pH of the Cd(II) solution (below 5). LGBCs are highly protonated at lower solution pH bringing about strong electrostatic repulsion interactions with cations such as Cd(II), which is responsible for the relatively lower Cd(II) adsorption [41].

### 2.3. Adsorption Mechanism of LGBCs towards Cd(II)

#### 2.3.1. Adsorption Isotherm

To reveal the relationships between the adsorbed Cd(II) and the residual Cd(II) in the solution, two of the most typical adsorption isotherms regarding the Langmuir and Freundlich models were employed to describe the processes of Cd(II) adsorption on LGBCs (Equations (S1) and (S2)). The Langmuir model represents a monolayer adsorption process occurring onto a homogeneous surface without molecule interactions. The Freundlich model reveals that the adsorption enthalpy on the surface of the adsorbent is heterogeneously distributed and enhances with the increasing coverage [42]. The fitting results are presented in Appendix A and the calculated parameters can be found in Table 3.

It can be found that both the Langmuir and Freundlich models can be employed to describe the adsorption of Cd(II) on LGBCs according to the higher R^2^ values obtained (0.926~0.993). In view of the definition of the Langmuir and Freundlich models, it is deduced that the adsorption of Cd(II) on LGBCs subjects it to a complex process involving homogeneous adsorption interaction as well as heterogeneous adsorption interaction. The theoretical maximum adsorption capacity of Cd(II) adsorption calculated using the Langmuir model is 2.12, 3.26, 9.15, and 18.54 mg/g for LG-300C, LG-400C, LG-500C, and LG-600C, respectively, which follows an order of LG-300C < LG-400C < LG-500C < LG-600C. This is consistent with the results presented in Figure 6a. The higher theoretical maximum adsorption capacity of LGBCs towards Cd(II) (especially for LG-600C) is comparable with the biomass-based biochars reported previously, as shown in Table 4. Moreover, a dimensionless factor R_L_ ($R_{L}=\frac{1}{1+K_{l}C_{0}}$) can be calculated based on the K_l_ from the Langmuir model to predict the affinities between Cd(II) and LGBCs [43]. R_L_ > 1 and 1 > R_L_ > 0 suggests the adsorption to be unfavorable and favorable, respectively. The calculated values of R_L_ of LGBCs for Cd(II) adsorption are less than 1, indicating that the adsorption of Cd(II) on LGBCs is favorable.

#### 2.3.2. Adsorption Kinetics

The relationships between contact time and Cd(II) adsorption on LGBCs can be characterized by adsorption kinetics. The adsorption kinetic data was fitted using pseudo-first-order and pseudo-second-order models (Equations (S3) and (S4)). Particularly, the pseudo-first-order model corresponds to a reversible adsorption reaction while a chemisorption process can be confirmed with the pseudo-second-order model [34]. The fitting results are shown in Appendix A (kinetic curves) and Table 5 (kinetic parameters). The pseudo-first-order model can simulate the adsorption of Cd(II) onto LGBCs with relatively higher R^2^ values ranging from 0.978 to 0.998, which are slightly higher than those of the pseudo-second-order model in the range of 0.930~0.980. In view of the calculated equilibrium adsorption capacity (Q_e,the_), the Q_e,the_ from the pseudo-first-order model are 0.91, 1.08, 5.94, and 8.12 mg/g for LG-300C, LG-400C, LG-500C, and LG-600C, respectively, which are much closer to the experimental data (Q_e,exp_) of 0.90, 1.08, 5.98, and 8.06 mg/g in comparison to those from the pseudo-second-order model. It is thus suggested that the pseudo-first-order model is a relatively ideal model to describe the Cd(II) adsorption on LGBCs. Most studies on the adsorption of Cd(II) by biochars present that the most ideal kinetic model is the pseudo-second-order model [41,44]. The differences observed in the most ideal kinetic model reveal that the mechanism of Cd(II) adsorption by LGBCs may be different from those reported to some extent. It is worth noting that the R^2^ fitted using the pseudo-second-order model are also within an acceptable range (>0.90), which may indicate that the adsorption of Cd(II) by LGBCs can also be described with a pseudo-second-order model. In other words, according to the definition of the pseudo-second-order model, the adsorption of Cd(II) by LGBCs is mainly a chemically driven processes.

#### 2.3.3. Thermodynamic Analysis

The calculated thermodynamic parameters according to Equations (S5)–(S10) for the adsorption of Cd(II) on LGBCs including ΔG^0^, ΔH^0^, and ΔS^0^ are presented in Table 6. Among them, ΔG^0^ can be employed to reveal the spontaneity of the adsorption reaction (a negative ΔG^0^ value and a positive ΔG^0^ value separately assigned to a spontaneous reaction and an unspontaneous reaction), ΔH^0^ indicates the change in system energy, and ΔS^0^ is related to the random state in the adsorption process and the affinity between the adsorbent and the adsorbate [43]. The values of ΔG^0^ for LGBCs at different ambient temperatures are negative (−2.23~−15.83 kJ/mol), indicating that the adsorption of Cd(II) by LGBCs are spontaneous reactions. Moreover, the decrease in ΔG^0^ values with the increase in ambient temperature suggest that increasing ambient temperature favors the adsorption of Cd(II). Additionally, the increase in pyrolysis temperature also leads to a reduction in the ΔG^0^ values, demonstrating that LGBCs prepared at higher pyrolysis temperatures are more favorable for the adsorption of Cd(II). Similar findings can be readily found in previous studies [35]. The positive ΔH^0^ values of 42.94~53.53 kJ/mol suggest endothermic reactions for Cd(II) adsorption by LGBCs, indicating that the increasing ambient temperature would be in favor of the Cd(II) adsorption. This coincided with the experimental data presented in Figure 6c.

#### 2.3.4. Potential Mechanisms

The unique physicochemical properties of LGBCs determine their adsorption capability as well as potential mechanisms for Cd(II) adsorption. As mentioned above, with the increase in pyrolysis temperature, the surface roughness of LGBCs increases. Additionally, LGBCs obtained at a higher pyrolysis temperature have a larger specific surface area. However, the results of adsorption isotherms, kinetics, and thermodynamics indicate that the adsorption of Cd(II) by LGBCs is a spontaneous, chemically driven, and complex process. In other words, surface deposition based on the surface roughness of LGBCs and pore filling based on the pore structure of LGBCs are not the main mechanisms for the adsorption of Cd(II) by LGBCs [18,34]. On the contrary, the abundant surface functional groups (such as -OH, C-O-C, C=O, and C=C) and relatively high content of minerals exposed on the surface of LGBCs may play a significant role in the adsorption process. It is reported that oxygen-containing functional groups can efficiently adsorb heavy metal ions through complexation interaction, while cation-π electron interaction is the crucial mechanism for the adsorption of heavy metal ions on aromatic C=C [8]. FTIR results indicate that the oxygen-containing functional groups decrease as the pyrolysis temperature increases, while aromatic C=C gradually strengthens. This suggests that aromatic C=C can be an important active adsorption site, promoting the adsorption of Cd(II) by LGBCs at higher pyrolysis temperatures. Although the oxygen-containing functional groups of LGBCs obtained at higher pyrolysis temperatures decrease, the significantly increased specific surface area may enhance the total exposed oxygen-containing functional groups, thereby making a significant contribution to the adsorption of Cd(II).

On the other hand, the minerals on the biochar’s surface can adsorb heavy metal ions through ion exchange, precipitation, and other mechanisms [15,35]. To confirm the role of minerals in the adsorption process, alkali metal and alkaline earth metal ions (K^+^, Na^+^, Ca^2+^, Mg^2+^) in the solution after Cd(II) adsorption by LGBCs were detected, while water-treated samples were used as controls (see Table 7). The results show that there are relatively few water-soluble metal ions in LGBCs, and their quantity decreases with increasing pyrolysis temperature. This is mainly attributed to the formation of more stable mineral products through interactions between minerals at higher pyrolysis temperatures. However, the analysis of metals in the equilibrium solution after Cd(II) adsorption by LGBCs reveals that a considerable amount of metal is released into the solution, obviously surpassing the amount of water-soluble metals. This indicates that Cd(II) can be efficiently adsorbed by LGBCs through the release of metal ions. It is worth noting that, although LGBCs obtained at higher pyrolysis temperatures release more metals, the proportion of metals released compared to the total adsorption capacity of LGBCs decreases rapidly from 36.66% for LG-300C to 5.65% for LG-600C. This suggests that both surface functional groups and minerals of LGBCs contribute to the adsorption of Cd(II), but the contribution of surface functional groups is greater.

#### 2.3.5. Reusability of LGBCs

The reusability (adsorption–desorption) is an important reference for assessing the applications and costs of adsorbents [50,51]. The above study shows that LGBCs (especially LG-600C) have excellent performance in treating heavy-metal-contaminated wastewater. Therefore, taking LG-600C as an example, the reusability of LGBCs is evaluated (see Figure 7). The results show that, after five adsorption–desorption cycles, LG-600C still maintains a high removal rate of Cd(II) (83.47%). Meanwhile, using EDTA as the de-sorbent, the desorption efficiency of LG-600C remains above 90%. Accordingly, it can be confirmed that LG-600C has excellent reusability for the treatment of heavy-metal-contaminated wastewater.

## 3. Materials and Methods

### 3.1. Materials

Lignin (LG) was extracted from black liquor from a papermaking mill in Hunan Province using an acid precipitate method, as described in our previous study [52], and cadmium nitrate (Cd(NO)_3_·4H_2_O) was purchased from Sigma-Aldrich (Burlington, MA, USA) with analytical grade.

### 3.2. Preparation of LGBCs

The preparation of biochar was carried out in a vertical fixed bed, as shown in Appendix A. A total of 1 g of sample (LG) was weighed and placed at the sample position. N_2_ was charged as a protective gas with a flow rate of 0.1 m^3^/h. The temperature was raised to the target temperature at a heating rate of 10 °C/min as the air in the device was exhausted. After 2 h at the target temperature, the device was allowed to cool down naturally to room temperature. The residues were collected, which were assigned to lignin-based biochars (LGBCs). Particularly, the target temperature was controlled at 300~600 °C. LGBCs prepared at different temperatures were named LG-XC, where X was the corresponding preparation temperature.

### 3.3. Characterization of LGBCs

The microstructures of samples were observed using a scanning electron microscope (SEM) (Inspect F50, FEI, Hillsboro, OR, USA). N_2_ adsorption/desorption was employed to analyze the pore structure of samples (Nova 2200e, Quantachrome, Boynton Beach, FL, USA). The specific surface area of samples was calculated using the Brunauer–Emmett–Teller (BET) method with relative pressure ranging from 0.05 to 0.3. The total pore volume was determined based on the nitrogen capacity at the highest relative pressure, while the average pore size was calculated using the Barrett–Joyner–Halenda (BJH) method. The mineral components of the samples were analyzed using an inductively coupled plasma emission spectrometer following treatment with a mixed acid solution composed of nitric and hydrofluoric acids (Agilent, 720, Santa Clara, CA, USA). The surface functional groups of samples were analyzed using a Fourier transform infrared spectrometer (Nicolet 6700, Thermo Fisher Scientific, Waltham, MA, USA). The pyrolysis behavior and thermal stability of samples were analyzed using a thermogravimetric analyzer (TGA) (TG209 F3, Netzsch, Selb, Germany). The crystalline substances and crystal structure in the samples were determined with a powder X-ray diffractometer (XRD) using a Cu target as the radiation source (wavelength 1.5406 Å) under an operating voltage and current of 40 kV and 40 mA, respectively (SmartLAB 3, Rigaku, Tokyo, Japan). Raman spectroscopy (inVia, Renishaw, Wotton-under-Edge, UK) was used to analyze the content of amorphous and graphitized components in the samples. The elemental composition of samples including C, H, O, N, and S (O calculated by difference) were determined using an elemental analyzer (EA112, Thermo Finnigan, San Jose, CA, USA), while the proximate analysis was measured using a horizontal tube furnace (OTL 1200, Nanjing Nanda, Nanjing, China). For pH determination, the samples were mixed with distilled water in a mass/volume ratio of 1:20.

### 3.4. Adsorption Tests of LGBCs

A precise quantity of LGBCs was weighed and deposited into a 100 mL conical flask which contained 50 mL of Cd(II) solution. This flask was then agitated in a shaker at a velocity of 120 rpm. Following the completion of the reaction, the suspension in the flask was evaluated using an atomic absorption spectrometry to identify the remaining Cd(II) concentration (FAAS-M6, Thermo, Waltham, MA, USA). Concurrently, the concentration of alkali and alkaline earth metal ions including K(I), Na(I), Ca(II), and Mg(II) in the suspension was also determined using an inductively coupled plasma emission spectrometer. Experiments using 50 mL of deionized water instead of Cd(II) solution were conducted following the same procedure for control purposes. The initial concentration of Cd(II), contact time, ambient temperature, and solution pH were set as follows: 5~60 mg/L, 2 min~8 h, 15~45 °C, and 1~5, respectively.

After adsorbing Cd(II), LG-600C was added to 20 mL of 0.1mol/L EDTA solution and desorbed for 30 min in an ultrasonic device. The suspension after ultrasonic treatment was then centrifuged in a high speed centrifuge, and the concentration of Cd(II) in the filtrate was analyzed using atomic absorption spectrometry to calculate the desorption efficiency. Meanwhile, the LG-600C at the bottom of the centrifuge tube was dried. The dried LG-600C was used for a new round of adsorption–desorption experiments for a total of 5 cycles.

## 4. Conclusions

SEM analysis reveals that, as the pyrolysis temperature increases, the surface roughness and pore structure of LGBCs also increase. Specifically, the specific surface area rises from 3.14 m^2^/g for LG-300C to 89.76 m^2^/g for LG-600C. At higher pyrolysis temperatures, the presence of surface functional groups in LGBCs, particularly O/S-containing groups, decreases significantly. LGBCs contain a relatively abundant amount of minerals, and the mineral content increases with the pyrolysis temperature. The adsorption of Cd(II) by LGBCs is greatly influenced by the initial concentration of Cd(II), contact time, ambient temperature, and solution pH. The fitting results of adsorption isotherms, kinetics, and thermodynamics models suggest that the adsorption of Cd(II) by LGBCs follows a spontaneous, chemically dominant, and complex process. According to the Langmuir adsorption isotherm model, the theoretical maximum adsorption capacities of LG-300C, LG-400C, LG-500C, and LG-600C for Cd(II) are calculated to be 2.12, 3.26, 9.15, and 18.54 mg/g, respectively, which are comparable to biomass-based biochar reported in the literature (especially LG-600C). Analysis of the adsorption mechanism reveals that the main mechanism for the adsorption of Cd(II) by LGBCs is the complexation interactions caused by surface functional groups, followed by precipitation/ion exchange interactions related to minerals.

## Figures and Tables

**Figure 1 molecules-28-07694-f001:**
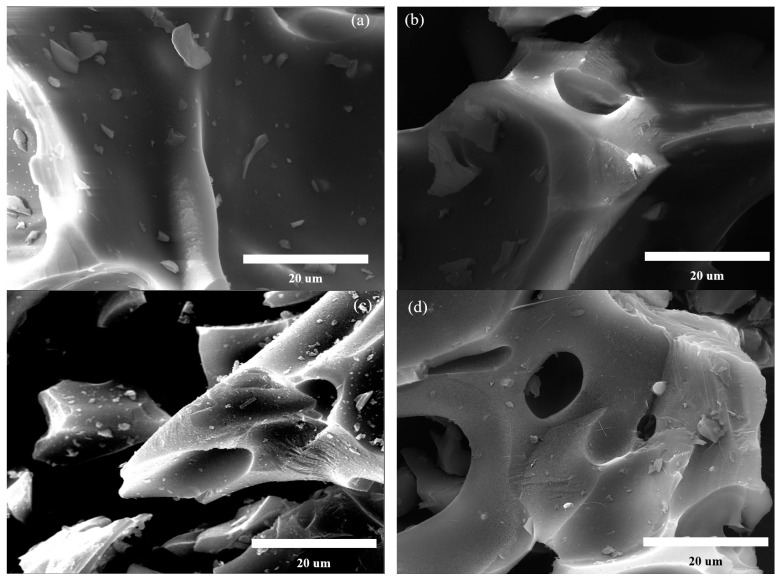
SEM images of LGBCs: (**a**) LG-300C, (**b**) LG-400C, (**c**) LG-500C, (**d**) LG-600C.

**Figure 2 molecules-28-07694-f002:**
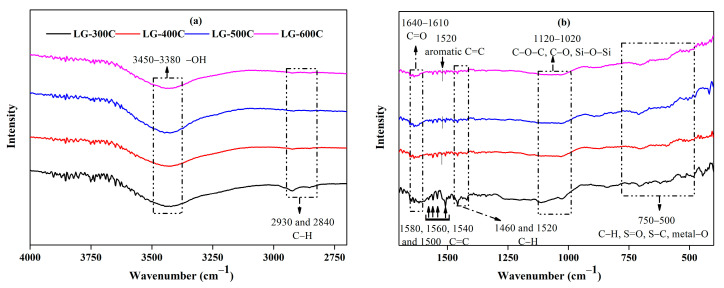
FT-IR spectra of LGBCs. (**a**) Wavenumber in the range of 4000–2700 cm^−1^. (**b**) Wavenumber in the range of 1700–400 cm^−1^.

**Figure 3 molecules-28-07694-f003:**
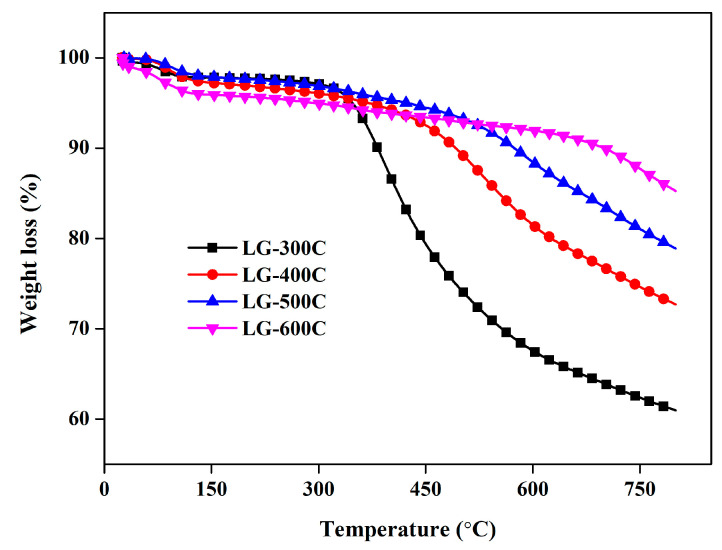
Thermogravimetric analysis of LGBCs.

**Figure 4 molecules-28-07694-f004:**
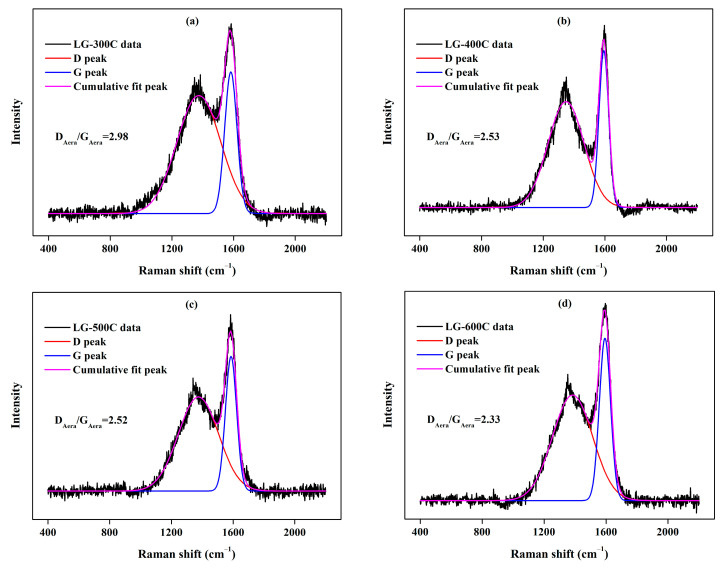
Raman spectra of LGBCs. (**a**) LG-300C data and peak fitting, (**b**) LG-400C data and peak fitting, (**c**) LG-500C data and peak fitting, (**d**) LG-500C data and peak fitting.

**Figure 5 molecules-28-07694-f005:**
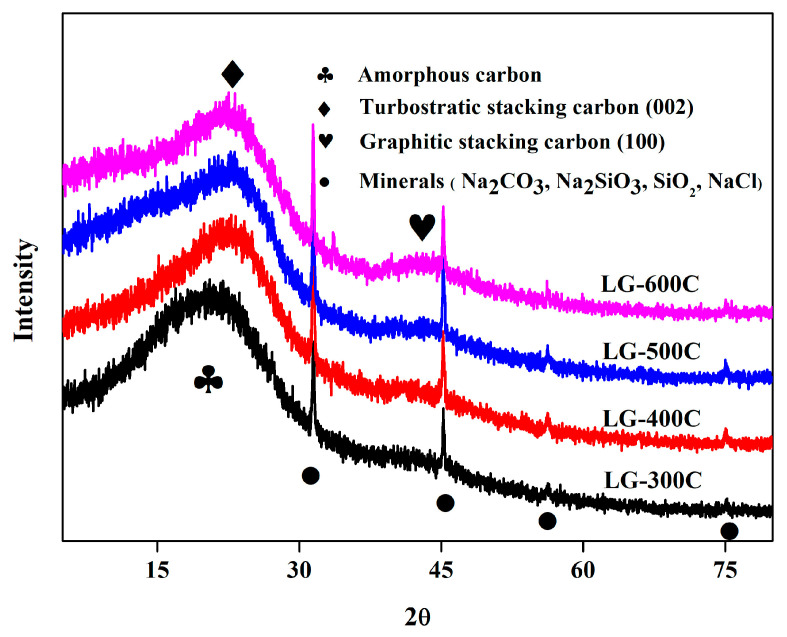
XRD patterns of LGBCs.

**Figure 6 molecules-28-07694-f006:**
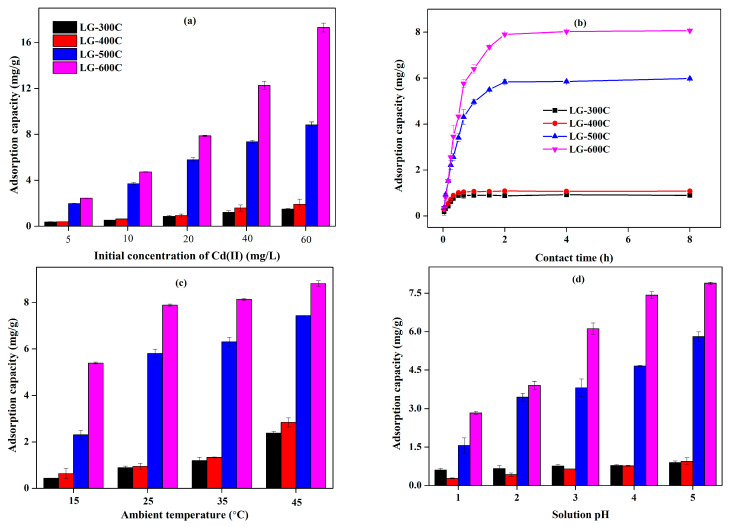
Effects of the operating conditions of the adsorption capability of LGBCs towards Cd(II). (**a**) Effect of initial concentration of Cd(II). (**b**) Effect of contact time. (**c**) Effect of ambient temperature. (**d**) Effect of solution pH. The initial concentration of Cd(II), contact time, ambient temperature, and solution pH were set as 5~60 mg/L, 2 min~8 h, 15~45 °C, and 1~5, respectively.

**Figure 7 molecules-28-07694-f007:**
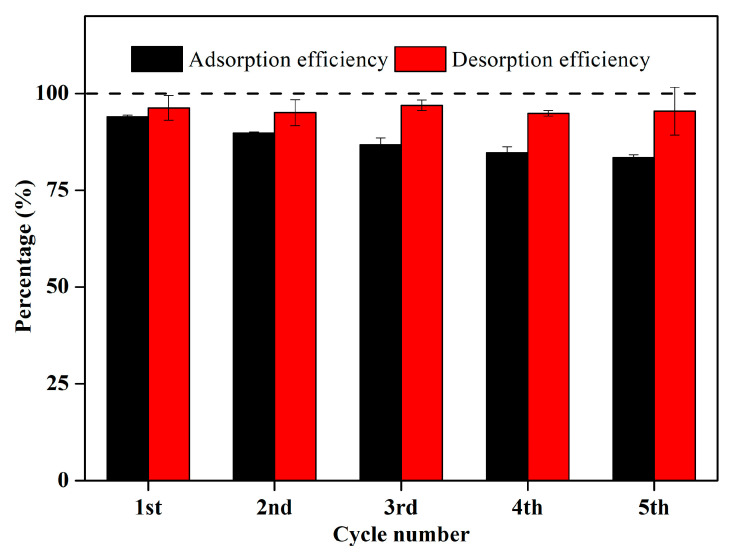
Reusability of LG-600C after five successive adsorption–desorption cycles. The initial concentration of Cd(II), contact time, ambient temperature, and solution pH were set as follows: 10 mg/L, 8 h, 25 °C, and 5, respectively.

**Table 1 molecules-28-07694-t001:** Physicochemical properties of LGBCs.

	Specific Surface Area (m^2^/g)	Micropore Specific Surface Area (m^2^/g)	Total Pore Volume (m^3^/g)	Micropore Pore Volume (m^3^/g)	Average Pore Size (nm)	Elemental Content (%)				Atomic Ratio	
						C	H	O	S	H/C	O/C
LG-300C	3.14	1.07	0.005	0.004	64.73	60.21	5.48	29.44	1.45	0.09	0.49
LG-400C	8.42	1.26	0.006	0.004	25.68	67.08	4.82	23.04	1.02	0.07	0.34
LG-500C	67.32	25.80	0.12	0.04	14.15	73.41	4.20	16.38	0.47	0.06	0.22
LG-600C	89.76	44.77	0.15	0.09	5.36	77.68	3.99	11.75	0.36	0.05	0.15

**Table 2 molecules-28-07694-t002:** Proximate analysis, mineral components, and other properties of LGBCs.

	pH	pH_pzc_	Proximate Analysis			Minerals				
			FC ^a^	VM ^b^	Ash	K	Na	Ca	Mg	Cd
			%			mmol/kg				
LG-300C	5.98	5.08	29.99	64.49	3.42	4.56	44.67	0.42	1.78	- ^c^
LG-400C	6.84	5.92	39.78	54.22	4.04	6.98	51.87	0.98	1.98	-
LG-500C	7.34	6.76	46.32	46.38	5.54	7.76	53.06	1.87	2.28	-
LG-600C	7.87	6.96	71.67	21.08	6.22	9.22	65.09	2.25	3.69	-

^a^ and ^b^ are fixed carbon and volatile matter, respectively; ^c^ represents under detection limitation.

**Table 3 molecules-28-07694-t003:** Adsorption isotherm parameters of Cd(II) adsorption on LGBCs.

Sample	Langmuir Model			Freundlich Model		
	Q_m_ (mg/g)	K_l_ (L/mg)	R^2^	K_f_ (mg^(1−n)^L^n^/g)	n	R^2^
LG-300C	2.12	0.04	0.984	0.17	1.88	0.993
LG-400C	3.26	0.03	0.985	0.16	1.63	0.992
LG-500C	9.15	0.23	0.971	2.55	2.99	0.965
LG-600C	18.54	0.21	0.926	4.79	2.65	0.967

**Table 4 molecules-28-07694-t004:** Comparison of the maximum adsorption capacity of Cu(II) with various biochar-related adsorbents.

No.	Biochar Sources	Prepared Conditions	Adsorption Conditions	Adsorption Capacity (mg/g)	References
1	Pleurotus ostreatus spent substrate	Pyrolysis at 500 °C for 2 h	Initial concentration in the range of 10~300 mg/L at 25 °C with solution pH of 7 for 24 h	12.63	[44]
2	Rice husk	Pyrolysis at 500 °C for 2 h	Initial concentration in the range of 20~800 mg/L at 25 °C for 24 h	5.55	[45]
3	Corn stalk	Pyrolysis at 600 °C for 2 h	Initial concentration in the range of 5~50 mg/L at 25 °C with solution pH of 6 for 24 h	7.02	[46]
4	Rice straw	Pyrolysis at 400 °C for 2 h	Initial concentration in the range of 1~60 mg/L at 25 °C with solution pH of 6 for 24 h	10.07	[47]
5	Giant Miscanthus	Pyrolysis at 600 °C for 1 h	Initial concentration in the range of 1~50 mg/L at 25 °C with solution pH of 7 for 48 h	12.96	[48]
6	Wheat straw	Hydrothermal carbonization at 600 °C for 4 h	Initial concentration in the range of 10~100 mg/L for 24 h	1.70	[49]
7	Black liquor lignin	Pyrolysis at 600 °C for 2 h	Initial concentration in the range of 5~60 mg/L at 25 °C with solution pH of 5 for 8 h	18.54	Present work

**Table 5 molecules-28-07694-t005:** Adsorption kinetic parameters of Cd(II) adsorption on LGBCs.

Sample	Q_e,exp_/(mg/g)	Pseudo-First-Order Kinetics			Pseudo-Second-Order Kinetics		
		k_1_ (1/h)	Q_e_ (mg/g)	R^2^	k_2_ (g/mg h)	Q_e_ (mg/g)	R^2^
LG-300C	0.90	4.84	0.91	0.979	7.26	0.99	0.930
LG-400C	1.08	4.84	1.08	0.978	6.26	1.16	0.948
LG-500C	5.98	1.80	5.94	0.998	0.32	6.76	0.980
LG-600C	8.06	1.59	8.12	0.995	0.20	9.35	0.972

**Table 6 molecules-28-07694-t006:** Adsorption thermodynamic parameters of Cd(II) adsorption on LGBCs.

Sample	ΔG^0^(kJ/mol)				ΔH^0^/kJ/mol	ΔS^0^/J/(mol·K)
	288 K	298 K	308 K	318 K		
LG-300C	−2.23	−4.20	−5.18	−7.55	46.49	169.26
LG-400C	−3.19	−4.36	−5.50	−8.18	42.94	159.21
LG-500C	−6.74	−10.75	−11.64	−13.40	53.53	170.19
LG-600C	−9.99	−13.18	−14.02	−15.83	43.00	185.64

**Table 7 molecules-28-07694-t007:** Exchanged ions during adsorption process.

	2K(I)	2Na(I)	Ca(II)	Mg(II)	TEC ^a^	AC ^b^	TEC/AC(%) ^c^
	mmol/kg						
LG-300C-Cd	0.66	6.21	0.04	0.03	6.94	18.93	36.66
LG-300C-H_2_O	0.15	0.85	- ^d^	0.01	1.01	/ ^e^	/
LG-400C-Cd	0.76	5.98	0.04	0.05	6.83	29.11	23.46
LG-400C-H_2_O	0.10	0.62	0.01	0.01	0.74	/	/
LG-500C-Cd	0.78	7.83	0.08	0.07	8.76	81.70	10.72
LG-500C-H_2_O	0.02	0.65	0.01	-	0.68	/	/
LG-600C-Cd	1.26	8.02	0.07	0.34	9.35	165.54	5.65
LG-600C-H_2_O	0.02	0.31	-	0.02	0.35	/	/

^a^ is the total exchanged capacity of alkali and alkaline metal ions (TEC = 2K(I) + 2Na(I) + Ca(II) + Mg(II)); ^b^ is the adsorption capacity of the biochar towards Cd(II) (the initial concentration, contact time, ambient temperature, and solution pH were 20 mg/L, 8 h, 25 °C, and 5, respectively); ^c^ is the ratio of TEC to AC; ^d^ indicates the value is below the detection limitation; ^e^ indicates this item is not applicable.

## Data Availability

The data presented in this study are available on request from the corresponding author.

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
