# Peer review of "The Physicochemical Characteristics and Heavy Metal Retention Capability of Black Liquor Lignin-Based Biochars"

_molecules, 2023, doi:10.3390/molecules28237694_

Round 1
Reviewer 1 Report
Comments and Suggestions for Authors
The authors present a potentially good work, performed experimental work systematically. The appropriate techniques are always welcome. This paper is worthy of publication in this journal after a minor revision.
1. There are many factors affect the adsorption properties, such as the concentration of humic acid, ion species and ionic strength. Can you add these parts in you manuscript?
2. As the reuse is an important property of the material, can this material be reused?
3. Please add the experiment condition on the Fig. 4, Fig. 5 and Fig. 6.
4. please unify the formats of all references, for example reference 9,15,24,27.
5.There are some formatting errors, such as incorrect parentheses and units, the formatting of the full text needs to be checked.
Author Response
The authors present a potentially good work, performed experimental work systematically. The appropriate techniques are always welcome. This paper is worthy of publication in this journal after a minor revision.
Many thanks for your comments on our manuscript. Based on your suggestions, we have carefully revised the manuscript as possible as we can. We hope the revised version could satisfy the requirement of Molecules. Please check the yellow parts in the revised manuscript.
Q1: There are many factors affect the adsorption properties, such as the concentration of humic acid, ion species and ionic strength. Can you add these parts in you manuscript?
Answer: Indeed, as you've mentioned, a multitude of factors influence the adsorption process. Beyond the initial concentration of Cd(II), contact time, ambient temperature and solution pH investigated in our study, variables such as dosage, coexisting ions, the concentration of humic acid, ion species and ionic strength, among others, would play a role. At present, our research is focused on an initial assessment of LGBCs' efficacy in treating wastewater. Taking your advice into account, we will conduct more systematic research in the future and report the findings then. Therefore, we are very sorry for not fully adopting your suggestions.
Q2: As the reuse is an important property of the material, can this material be reused?
Answer: Based on your suggestion, we have supplemented the experiment to evaluate the reusability of LG-600C. Please check the 3.4 Adsorption tests of LGBCs in the revised manuscript.
After adsorbing Cd(II), LG-600C was added to 20 mL of 0.1mol/L EDTA solution and desorbed for 30 minutes in an ultrasonic device. The suspension after ultrasonic treatment was then centrifuged in a high speed centrifuge, and the concentration of Cd(II) in the filtrate was analyzed using an atomic absorption spectrometry to calculate the desorption efficiency. Meanwhile, the LG-600C at the bottom of the centrifuge tube was dried. The dried LG-600C was used for a new round of adsorption-desorption experiment for a total of 5 cycles.
At the same time, the results of the reusability of LG-600C were shown in Figure 8.
2.3.5 Reusability of LG-600C
The reusability (adsorption-desorption) is an important reference for assessing the applications and costs of adsorbents [1,2]. The above study shows that LGBCs (especially LG-600C) have excellent performance in treating heavy metal-contaminated wastewater. Therefore, taking LG-600C as an example, the reusability of LGBCs is evaluated (see Figure 8). The results show that after five adsorption-desorption cycles, LG-600C still maintains a high removal rate of Cd(II) (83.47%). Meanwhile, using EDTA as the de-sorbent, the desorption efficiency of LG-600C remains above 90%. Accordingly, it can be well confirmed that LG-600C has excellent reusability for the treatment of heavy metal-contaminated wastewater.
Figure 8. Reusability of LG-600C after 5 successive adsorption-desorption cycles.
References:
- Kim, H.-S.; Choi, H.-J., Design of a Novel Sericite–Phosphoric Acid Framework for Enhancement of Pb(II) Adsorption. Molecules 2023, 28, (21), 7395.
- Zhang, J.; Ren, H.; Fan, H.; Zhou, S.; Huang, J., One-Step Fabrication of Recyclable Konjac Glucomannan-Based Magnetic Nanoparticles for Highly Efficient Cr (VI) Adsorption. Molecules 2023, 28, (20), 7100.
Q3: Please add the experiment condition on the Fig. 4, Fig. 5 and Fig. 6.
Answer: Based on your suggestion, we have added the experimental conditions to the relevant Figures. Please check the Figure 5-8 in the revised manuscript.
Q4: Please unify the formats of all references, for example reference 9,15,24,27.
Answer: Based on your suggestion, we have unified the formats of the references. Please check the revised manuscript.
Q5: There are some formatting errors, such as incorrect parentheses and units, the formatting of the full text needs to be checked.
Answer: Based on your suggestion, the formatting of the full text has been seriously checked and revised. Please check the revised manuscript.

Reviewer 2 Report
Comments and Suggestions for Authors
Manuscript ID: molecules-2688801
Title: Physicochemical characteristics and heavy metal retention capability of black liquor lignin-based biochars
Authors report on the annealed lignin produced from black liquor. Materials' structure, absorption kinetics and Cd(II) retention are rigorously analyzed. The application of the woodwork by-product to the contamination removal is of interest. However, the discussion of Raman and FTIR spectra has a considerable room for improvement, and some experimental details can be expanded as well. Therefore, I recommend a major revision of the manuscript.
Major comments:
1) Lines 141-142, "The stronger aromatic C=C peak observed in the higher pyrolysis temperature-derived LGBCs (such as LG-600C)": from Fig. 2(a), it is unclear if such a change in FTIR spectra takes place. Consider showing magnified images with the changing regions of the spectra to prove your point.
2) Could you comment on almost complete disappearance of C-H-related FTIR lines at ~2900 and 1420 cm-1 with 300°→400° annealing temperature increase? It doesn't match the elemental analysis data presented in Table 1, which shows a gradual decrease of the hydrogen content with 300°→600° temperature enlargement. Does C-H disappearance from FTIR mean that C-H bonds break almost completely at 400°, and the only hydrogen left is molecular hydrogen trapped in the structure?
3) I consider the 1590 cm-1 line assignment to C-O or C=O highly unconventional, as 1580-1600 cm-1 is a typical position of C=C bonds vibration [10.1016/j.tsf.2021.138993, 10.1080/10408347.2016.1157013]. Consider revising the presented interpretations.
4) Additionally, please revise the data related to C-O bonding assignment to the lines located in the vicinity of ~1600 cm-1. As shown in [10.1080/10408347.2016.1157013], Table 5, the vibrations at 1040-1415 cm-1 are related to C-O bonding, while 1600-1700 cm-1 region is more typical for C=O.
5) Additionally, as one can see from [10.1080/10408347.2016.1157013], a variety of lines can be detected in 500-750 cm-1 region (for example, C=O from -COOH group, see Table 5) -- why have authors attributed this region to hydrogen only?
6) Why LG-300C Raman spectra weren't detected? Is it related to Raman spectroscopy predominant sensitivity to the sp2-hybridized carbon? Have authors tried to change the excitation wavelength to a higher wavenumber to reduce the luminescent background overlapping the peaks? I also suppose that the feature observed at ~2300 cm-1 is related to the degradation of the sample under the excitation laser beam, as it shows that after diffraction grating shift the second region of the spectra was less intensive than the first one. In my opinion, acquiring less noisy spectra on different wavelength with lower excitation power would be beneficial for the discussion.
7) Lines 176-177, "the increased graphitization is related to the enrichment of aromatic C=C at higher pyrolysis temperature", lines 349-351, " FTIR results indicate that ... aromatic C=C gradually strengthens.". In my opinion, neither FTIR nor Raman spectroscopy demonstrate the graphitization of the material. In FTIR, one would expect ~1600 cm-1 line intensity to grow with the graphitization, while in Raman spectroscopy, G-line position is sensitive to the sp2/sp3 ratio, while "area D to area G" increase indicates the rise of the disorder of the material [10.1007/s00339-022-06062-2]. Please revise the presented data to verify if a considerable graphitization really takes place.
8) Elemental analysis presented in Table 1 shows the presence of C,N,O,S only, while elemental analysis of the same materials presented in Table 2 indicates the presence of K, Na, Ca and Mg. How is that possible?
9) Although authors claim that minerals are crucial for the films’ contamination removal performance, the background of their emergence remains obscured. What is the origin of minerals presence in the annealed samples? Are they present in the precursor lignin?
10) Lines 196-197, "To quantitatively analyze the content of mineral-related active adsorption sites, proximate analysis (was carried out)": please elaborate on the obtained results. What is the interplay between the fixed carbon/volatile matter/ash and mineral-related absorption sites? How can the fixed carbon/volatile matter/ash variation with annealing temperature be explained?
Minor comments:
11) In Section 2.1.2, please discuss the origin of sulphur present in the samples. How does its presence affect the metal retention properties of the material? Is sulphur present in the precursor lignin?
12) Line 86, "analysis shown in Fig. S2": figure S2 is referenced prior to fig. S1. Please revise.
13) Section 3.3, Raman studies: what excitation wavelength was used, what laser power was set, what was the auisition time, what diffraction grating was used?
14) Section 3.3, XRD studies: what X-ray wavelength was used?
15) Consider removing the luminescent background from the Raman spectra (Fig. 3a), re-normalizing them to the similar intensity and to show the spectra fitting to show that the spectra can be properly fitted by 2 lines. What type of lineshapes (Gauss/Lorentz) were used?
16) Table 2: how was mineral component analysis carried out?
17) Lines 329-331, "indicating that the increasing ambient temperature would be in favor of the Cd(II) adsorption. This is well coincided with the experiment data presented in Fig. 4b": Fig. 4b doesn't show the adsorption kinetics dependence on ambient temperature. Could you elaborate on what have you meant?
18) Section 2.3.3: please discuss how ΔG0, ΔH0, ΔS0 values were obtained.
19) Line 341, what do you mean by "surface deposition"? Did you mean physical or chemical adsorption?
20) How was the ion exchange-related data (Table 7) obtained? Why some of the ions are doubled ("2K(I)", "2Na(I)")?
21) Table 7, what is "TEC/AC"?
Comments on the Quality of English LanguageLine 144, by "lone pair electrons", did you mean "lone-pair electrons"?
Author Response
Authors report on the annealed lignin produced from black liquor. Materials' structure, absorption kinetics and Cd(II) retention are rigorously analyzed. The application of the woodwork by-product to the contamination removal is of interest. However, the discussion of Raman and FTIR spectra has a considerable room for improvement, and some experimental details can be expanded as well. Therefore, I recommend a major revision of the manuscript.
Many thanks for your comments on our manuscript. Based on your suggestions, we have carefully revised the manuscript as possible as we can. We hope the revised version could satisfy the requirement of Molecules. Please check the yellow parts in the revised manuscript.
Q1: Lines 141-142, "The stronger aromatic C=C peak observed in the higher pyrolysis temperature-derived LGBCs (such as LG-600C)": from Fig. 2(a), it is unclear if such a change in FTIR spectra takes place. Consider showing magnified images with the changing regions of the spectra to prove your point.
Answer: Based on your suggestions, we've made the corresponding improvements. Please check Figure 2 in the revised manuscript.
Figure 2. FT-IR spectra of LGBCs.
Q2: Could you comment on almost complete disappearance of C-H-related FTIR lines at ~2900 and 1420 cm-1 with 300°→400° annealing temperature increase? It doesn't match the elemental analysis data presented in Table 1, which shows a gradual decrease of the hydrogen content with 300°→600° temperature enlargement. Does C-H disappearance from FTIR mean that C-H bonds break almost completely at 400°, and the only hydrogen left is molecular hydrogen trapped in the structure?
Answer: Thanks for your suggestion; our previous results indeed had some flaws. Based on that, we conducted another measurement of the sample, and the specific results can be found in Figure 2. According to the results, we observed that the C-H vibration peaks gradually weaken with the increase of pyrolysis temperature, rather than disappearing suddenly at a certain temperature. This finding is well consistent with previous research and also well matches the elemental analysis results [1, 2]. Please check Figure 2 in the revised manuscript.
References:
- Keiluweit, M.; Nico, P. S.; Johnson, M. G.; Kleber, M., Dynamic Molecular Structure of Plant Biomass-Derived Black Carbon (Biochar). Environmental Science & Technology 2010, 44, (4), 1247-1253.
- Wang, Z. H.; Guo, H. Y.; Shen, F.; Yang, G.; Zhang, Y. Z.; Zeng, Y. M.; Wang, L. L.; Xiao, H.; Deng, S. H., Biochar produced from oak sawdust by Lanthanum (La)-involved pyrolysis for adsorption of ammonium (NH4+), nitrate (NO3-), and phosphate (PO43-). Chemosphere 2015, 119, 646-653.
Q3: I consider the 1590 cm-1 line assignment to C-O or C=O highly unconventional, as 1580-1600 cm-1 is a typical position of C=C bonds vibration [10.1016/j.tsf.2021.138993, 10.1080/10408347.2016.1157013]. Consider revising the presented interpretations.
Answer: Thank you for your suggestion; our previous analysis did have some flaws. The references you provided are very helpful to us. The absorption peak at 1590 cm-1 is unlikely to be attributed to C=O. We have made revisions to the results (Figure 3) as well as corresponding descriptions. Please check Figure 2 and 2.1.2 Surface functional groups section in the revised manuscript.
Q4: Additionally, please revise the data related to C-O bonding assignment to the lines located in the vicinity of ~1600 cm-1. As shown in [10.1080/10408347.2016.1157013], Table 5, the vibrations at 1040-1415 cm-1 are related to C-O bonding, while 1600-1700 cm-1 region is more typical for C=O.
Answer: As mentioned in Q 3, we have made revisions to the results and the corresponding descriptions. Please check Figure 2 and 2.1.2 Surface functional groups section in the revised manuscript.
Q5: Additionally, as one can see from [10.1080/10408347.2016.1157013], a variety of lines can be detected in 500-750 cm-1 region (for example, C=O from -COOH group, see Table 5) -- why have authors attributed this region to hydrogen only?
Answer: Indeed, the range of 700-500 cm-1 belongs to the fingerprint region, which contains a very large variety of functional groups (such as C-H, S=O, S-C, metal-O) and is extremely complex. We hadn't considered functional groups other than C-H before, but now we have made revisions. Please check Figure 2 and 2.1.2 Surface functional groups section in the revised manuscript.
Q6: Why LG-300C Raman spectra weren't detected? Is it related to Raman spectroscopy predominant sensitivity to the sp2-hybridized carbon? Have authors tried to change the excitation wavelength to a higher wavenumber to reduce the luminescent background overlapping the peaks? I also suppose that the feature observed at ~2300 cm-1 is related to the degradation of the sample under the excitation laser beam, as it shows that after diffraction grating shift the second region of the spectra was less intensive than the first one. In my opinion, acquiring less noisy spectra on different wavelength with lower excitation power would be beneficial for the discussion.
Answer: Indeed, thanks to your suggestion, we re-measured the samples. The results confirmed that there were some flaws in our previous data. We have analyzed the newly obtained data. Please check Figure 4 and 2.1.2 Surface functional groups section in the revised manuscript.
|
|
|
Figure 4. Raman spectra of LGBCs.
Additionally, we found that compared to the 2D peak, the D peak and G peak are more commonly used to analyze the forms of carbon in biochars. Therefore, we mainly focused our analysis on the results of the D peak and G peak. Please check Figure 4 and 2.1.2 Surface functional groups section in the revised manuscript.
Q7: Lines 176-177, "the increased graphitization is related to the enrichment of aromatic C=C at higher pyrolysis temperature", lines 349-351, " FTIR results indicate that ... aromatic C=C gradually strengthens.". In my opinion, neither FTIR nor Raman spectroscopy demonstrate the graphitization of the material. In FTIR, one would expect ~1600 cm-1 line intensity to grow with the graphitization, while in Raman spectroscopy, G-line position is sensitive to the sp2/sp3 ratio, while "area D to area G" increase indicates the rise of the disorder of the material [10.1007/s00339-022-06062-2]. Please revise the presented data to verify if a considerable graphitization really takes place.
Answer: Indeed, neither FTIR nor Raman spectroscopy demonstrate the graphitization of the material. We've revised the results based on the latest data. Please check Figure 2 and 4 in the revised manuscript.
The ratio of D peak area and G peak area (DAera/GAera), usually being employed to reveal the defect degree of carbon materials, is calculated to be 1.56, 1.41, 1.26 and 1.18 for LG-300C, LG-400C, LG-500C and LG-600C, respectively. In other words, the DAera/GAera values of LGBCs decrease with the increasing pyrolysis temperature, indicating that structural defects reduce with the elevation of pyrolysis temperature. Combination with the results of FT-IR, the decrease of structural defects is mainly caused by the decomposition of O-containing surface functional groups at higher pyrolysis temperature.
Q8: Elemental analysis presented in Table 1 shows the presence of C,N,O,S only, while elemental analysis of the same materials presented in Table 2 indicates the presence of K, Na, Ca and Mg. How is that possible?
Answer: In fact, elemental analysis and mineral composition analysis are not conflicting but interrelated. Elemental analysis is carried out using gas chromatography-mass spectrometry. Elemental analysis employs GC-MS to quantify inorganic elements such as C, H, N, S, and others, with O being determined through a subtraction technique. To accurately assess inorganic elements, one must account for the ash content. Minerals, on the other hand, predominantly reside within this ash. Consequently, the metallic elements within the samples are quantified post-ashing.
Q9: Although authors claim that minerals are crucial for the films’ contamination removal performance, the background of their emergence remains obscured. What is the origin of minerals presence in the annealed samples? Are they present in the precursor lignin?
Answer: As described in the Materials and Methods section, the lignin used in this study came from black liquor. There was a large amount of minerals in the black liquor, mainly originating from the salts added during the paper pulping process. Most of the ash content was removed during the extraction of lignin from the black liquor and only a small portion was carried over into the extracted lignin. The ash in extracted lignin played an important role in the thermal conversion of lignin and its subsequent applications. Please check 3.1 Materials section the revised manuscript.
Q10: Lines 196-197, "To quantitatively analyze the content of mineral-related active adsorption sites, proximate analysis (was carried out)": please elaborate on the obtained results. What is the interplay between the fixed carbon/volatile matter/ash and mineral-related absorption sites? How can the fixed carbon/volatile matter/ash variation with annealing temperature be explained?
Answer: As described earlier, the minerals in the ash played an important role in the adsorption of Cd(II) by LGBCs. Understanding the ash through proximate analysis allowed for a certain degree of insight into the minerals' contribution to the Cd(II) adsorption by LGBCs. Fixed carbon, volatile matter, and ash content were closely related to the pyrolysis temperature. Generally, due to the continuous decomposition of organic components in the biochar precursor, the resulting carbon materials had increased fixed carbon and ash content at higher pyrolysis temperatures, while the volatile matter decreased. Please check the revised manuscript.
Q11: In Section 2.1.2, please discuss the origin of sulphur present in the samples. How does its presence affect the metal retention properties of the material? Is sulphur present in the precursor lignin?
Answer: The description for the origin of S in the sample has been added. The S content mainly came from the paper pulping process due to the addition of S-contained salts such as sodium benzenesulfonate and sodium sulfate. Meanwhile, the use of sulfuric acid to extract LG was also an important source of sulfur in LGBCs. Similar to O-containing surface functional groups, the presence of S can provide lone-pair electrons and serve as effectively active adsorption sites for heavy metal ions removal via the formation of complexes. Please check 2.1.2 Surface functional groups section in the revised manuscript.
Q12: Line 86, "analysis shown in Fig. S2": figure S2 is referenced prior to fig. S1. Please revise.
Answer: Thank you for your suggestion, this oversight has been revised. Please check the revised manuscript.
Q13: Section 3.3, Raman studies: what excitation wavelength was used, what laser power was set, what was the auisition time, what diffraction grating was used?
Answer: We're sorry about this. Our samples are usually sent to a certified testing company for analysis, and we're not really sure about the specific parameters. We only know that the excitation wavelength is 488 nm. We'll pay more attention to these details in the future.
Q14: Section 3.3, XRD studies: what X-ray wavelength was used?
Answer: The X-ray wavelength used in XRD test was 1.5406 Å. Based on your suggestion, we have supplemented this part of the information. Please check 3.3 Characterization of LGBCs section in the revised manuscript.
The crystalline substances and crystal structure in the samples were determined by a powder X-ray diffractometer (XRD) using a Cu target as the radiation source (wavelength 1.5406 Å) under an operating voltage and current of 40 kV and 40 mA, respectively (SmartLAB 3, Rigaku, Japan).
Q15: Consider removing the luminescent background from the Raman spectra (Fig. 3a), re-normalizing them to the similar intensity and to show the spectra fitting to show that the spectra can be properly fitted by 2 lines. What type of lineshapes (Gauss/Lorentz) were used?
Answer: Based on your suggestion, we removed the luminescent background from Raman's data, and re-normalized and fitted it. The fitting results are shown in Figure 3. Please check Figure 3 in the revised manuscript.
Figure 3. Raman spectra of LGBCs.
Moreover, during the peak fitting process, we found that the Lorentz model fitted the Raman data better than the Gauss model, so all the results presented in Figure 3 were derived from the Lorentz model. Please check the revised manuscript.
Q16: Table 2: how was mineral component analysis carried out?
Answer: Thank you for your suggestion. The method for analyzing mineral components in the samples has now been added to the Materials and Methods section. Please check 3.3 Characterization of LGBCs in the revised manuscript.
Q17: Lines 329-331, "indicating that the increasing ambient temperature would be in favor of the Cd(II) adsorption. This is well coincided with the experiment data presented in Fig. 4b": Fig. 4b doesn't show the adsorption kinetics dependence on ambient temperature. Could you elaborate on what have you meant?
Answer: Thank you for your suggestion. There was a mistake in our previous manuscript, it's not Figure 4b it should be Figure 5c. We have already made the correction. Please check 2.3.3 Thermodynamic analysis section in the revised manuscript.
Q18: Section 2.3.3: please discuss how ΔG0, ΔH0, ΔS0 values were obtained.
Answer: In fact, the methods to calculate the thermodynamic parameters including ΔG0, ΔH0, ΔS0 has already been presented in the Supplementary materials (Eq. S5-S10). We didn't articulate this clearly before, but we've made modifications now. Please check 2.3.3 Thermodynamic analysis section in the revised manuscript.
Q19: Line 341, what do you mean by "surface deposition"? Did you mean physical or chemical adsorption?
Answer: Surface deposition is a relatively weak physical interaction that is directly related to the surface roughness of the material. The rougher the surface of the material, the more likely surface deposition will play an important role during the adsorption process [1] .
References:
- Su, D. S.; Chen, X.; Weinberg, G.; Klein-Hofmann, A.; Timpe, O.; Hamid, S. B. A.; Schlögl, R., Hierarchically Structured Carbon: Synthesis of Carbon Nanofibers Nested inside or Immobilized onto Modified Activated Carbon. Angewandte Chemie International Edition 2005, 44, (34), 5488-5492.
Q20: How was the ion exchange-related data (Table 7) obtained? Why some of the ions are doubled ("2K(I)", "2Na(I)")?
Answer: The method to obtain the ion exchange-related data has already been added to the Materials and Methods section. Please check 3.4 Adsorption tests of LGBCs in the revised manuscript.
“A precise quantity of LGBCs was weighed and deposited into a 100 mL conical flask, which contained 50 mL of Cd(II) solution. This flask was then agitated in a shaker at a velocity of 120 rpm. Following the completion of the reaction, the suspension in the flask was evaluated using atomic absorption spectrometry to identify the remaining Cd(II) concentration (FAAS-M6, Thermo, USA). Concurrently, the concentration of alkali and alkaline earth metal ions including K(I), Na(I), Ca(II), and Mg(II) in the suspension was also determined by an inductively coupled plasma emission spectrometer (Agilent, 720, USA). Experiments using 50 mL of deionized water instead of Cd(II) solution were conducted following the same procedure for control purposes.”
The data for K and Na is multiplied by 2 because Cd(II) is a divalent metal. According to the principle of ion exchange, monovalent metals need to be multiplied by 2 [1-2].
References:
- Qin, K.; Li, J. L.; Yang, W. C.; Wang, Z. H.; Zhang, H. Y., Role of minerals in mushroom residue on its adsorption capability to Cd(II) from aqueous solution. Chemosphere 2023, 324.
- Wang, Z.; Shen, F.; Shen, D.; Jiang, Y.; Xiao, R., Immobilization of Cu(2+) and Cd(2+) by earthworm manure derived biochar in acidic circumstance. J Environ Sci (China) 2017, 53, 293-300
Q21: Table 7, what is "TEC/AC"?
Answer: TEC/AC is the ratio of the total exchanged capacity of alkali and alkaline metal ions (TEC) and the adsorption capacity of the biochar towards Cd(II) (AC). Our previous description was unclear, we have now improved and revised it. Please check Table 7 in the revised manuscript.
Table 7. Exchanged ions during adsorption process.
|
|
2K(I) |
2Na(I) |
Ca(II) |
Mg(II) |
TECa |
ACb |
TEC/AC(%)c |
|
mmol/kg |
|||||||
|
LG-300C-Cd |
0.66 |
6.21 |
0.04 |
0.03 |
6.94 |
18.93 |
36.66 |
|
LG-300C -H2O |
0.15 |
0.85 |
-d |
0.01 |
1.01 |
/e |
/ |
|
LG-400C -Cd |
0.76 |
5.98 |
0.04 |
0.05 |
6.83 |
29.11 |
23.46 |
|
LG-400C -H2O |
0.10 |
0.62 |
0.01 |
0.01 |
0.74 |
/ |
/ |
|
LG-500C -Cd |
0.78 |
7.83 |
0.08 |
0.07 |
8.76 |
81.70 |
10.72 |
|
LG-500C -H2O |
0.02 |
0.65 |
0.01 |
- |
0.68 |
/ |
/ |
|
LG-600C -Cd |
1.26 |
8.02 |
0.07 |
0.34 |
9.35 |
165.54 |
5.65 |
|
LG-600C -H2O |
0.02 |
0.31 |
- |
0.02 |
0.35 |
/ |
/ |
a is the total exchanged capacity of alkali and alkaline metal ions (TEC=2K(I)+2Na(I)+Ca(II)+Mg(II)); b is the adsorption capacity of the biochar towards Cd(II) (the initial concentration, contact time, ambient temperature and solution pH were 20 mg/L, 8 h, 25 ℃ and 5, respectively); c is the ratio of TEC to AC; d indicates the value is below the detection limitation; e indicates this item is not applicable.
Q22: Line 144, by "lone pair electrons", did you mean "lone-pair electrons"?
Answer: Yes, what we want to express is lone-pair electrons. Thank you for your suggestion, we have made the modification. Please check Line 138 in the revised manuscript.

Round 2
Reviewer 2 Report
Comments and Suggestions for Authors
Manuscript title: "Physicochemical characteristics and heavy metal retention capability of black liquor lignin-based biochars"
ID molecules-2688801
Revision 1.
Authors have answered most of the questons and addressed most of the iisues related to the manuscript. However, revised Raman spectra fitting raises some additional questions. Therefore, I recommend another major revision of the manuscript.
Major comment:
1) Fig. 4 indicates that Raman spectra can be properly fitted with 4 lines. However, the origin of 2 lines only (D and G-bands) is discussed in the text, indicating that 1200 cm-1 and 1500 cm-1 are obscured "fit peaks". Origin of all of the observed lines should be discussed and confirmed by other techniques of analysis.
As elemental analysis (Table 1) indicates that hydrogen is present in material structure, I suggest that the lines at ~1200 cm-1 and ~1500 cm-1 are related to the vibrations of C-C and C=C bonds of polyenic structures (conjugated polymer fragments which can originate from organic residuals) [10.3390/jcs7040156]. To verify if polyenes contribute to the Raman spectra, authors are encouraged to re-fit the spectrum of LG-500C (see comment 5) and compare the relative area of polyenic and graphitic stretching vibrations (polyenic C=C area to G-line) to see if its decrease trend matches the one of H/C ratio (Table 1).
Minor comments:
2) In the caption of Fig. 2 (line 141), (a) and (b) subfigures should be denoted.
3) Line 167, 169, notation in Fig. 4: by "(DAera/GAera)", did you mean "(DArea/GArea)"?
4) In the caption of Fig. 4 (line 174), (a-d) subfigures should be denoted.
5) In Fig. 4c, the line peaking at ~1500 cm-1 is ommited and peak fitting doesn't match the experimental data. Consider revising the fitting.
6) In Section 3.3, please denote what you mean by "ultimate analysis" (Table S1, lines 143 and 160 of the main body).
7) Conclusions, line 460, state that "the amount of aromatic C=C bonds increases", but the reply to Q7 indicates that "neither FTIR nor Raman spectroscopy demonstrate the graphitization of the material". Please revise the fragment.
8) Supplementary material, "Kf is the Freundlich affinity coefficient (mg(1-n)Ln/g)": if measurement units are denoted in the brackets, why does the notation contains dimensionless parameter "n"? What is Ln?
9) Supplementary material, section 1.3: please denote that the measurement units of Gibbs free energy, enthalpy, and entropy are Joules.
Author Response
Responses to the reviewers’ comments on “Physicochemical characteristics and heavy metal retention capability of black liquor lignin-based biochars” (Manuscript ID: Molecules-2688801).
Dear editor and reviewers:
We really appreciated you for your time and valuable suggestions on our manuscript. We have taken very careful considerations on your comments to improve our manuscript as possible as we can. In the revised manuscript, all the corrections and changes have been labeled in yellow. The detailed responses are as follows:
Reviewer #1:
Authors have answered most of the questons and addressed most of the iisues related to the manuscript. However, revised Raman spectra fitting raises some additional questions. Therefore, I recommend another major revision of the manuscript.Many thanks for your comments on our manuscript. Based on your suggestions, we have carefully revised the manuscript as possible as we can. We hope the revised version could satisfy the requirement of Molecules. Please check the yellow parts in the revised manuscript.
Q1: Fig. 4 indicates that Raman spectra can be properly fitted with 4 lines. However, the origin of 2 lines only (D and G-bands) is discussed in the text, indicating that 1200 cm-1 and 1500 cm-1 are obscured "fit peaks". Origin of all of the observed lines should be discussed and confirmed by other techniques of analysis.
Answer: Based on your previous suggestions and our reference literature review, we have adjusted the fitting of Raman spectra peaks from 4 to 2 (i.e., D peak and G peak). Accordingly, this issue can be avoided. In the future, we will further explore how to better analyze Raman data. Please check Figure 4 in the revised manuscript.
|
|
|
Figure 4. Raman spectra of LGBCs. (a) LG-300C data and peak fitting, (b) LG-400C data and peak fitting, (c) LG-500C data and peak fitting, (d) LG-500C data and peak fitting.
Q2: As elemental analysis (Table 1) indicates that hydrogen is present in material structure, I suggest that the lines at ~1200 cm-1 and ~1500 cm-1 are related to the vibrations of C-C and C=C bonds of polyenic structures (conjugated polymer fragments which can originate from organic residuals) [10.3390/jcs7040156]. To verify if polyenes contribute to the Raman spectra, authors are encouraged to re-fit the spectrum of LG-500C (see comment 5) and compare the relative area of polyenic and graphitic stretching vibrations (polyenic C=C area to G-line) to see if its decrease trend matches the one of H/C ratio (Table 1).
Answer: As mentioned, we have re-fitted the data of Raman spectra, reducing the number of fitted peaks from 4 to 2. Accordingly, this issue can be avoided. In the future, we will further explore how to better analyze Raman data. Please check Figure 4 in the revised manuscript.
Q3: In the caption of Fig. 2 (line 141), (a) and (b) subfigures should be denoted.
Answer: Thank you for your suggestion. The subfigures have been denoted. Please check Figure 2 in the revised manuscript.
|
|
Figure 2. FT-IR spectra of LGBCs. (a) Wavenumber in the range of 4000-2700 cm-1, (b) Wavenumber in the range of 1700-400 cm-1.
Q4: Line 167, 169, notation in Fig. 4: by "(DAera/GAera)", did you mean "(DArea/GArea)"?
Answer: Yes, we have made modifications. Please check Lines 167 and 169 and Figure 4 in the revised manuscript.
Q5: In the caption of Fig. 4 (line 174), (a-d) subfigures should be denoted.
Answer: Thank you for your suggestion. The subfigures have been denoted. Please check Figure 4 in the revised manuscript.
Q6: In Fig. 4c, the line peaking at ~1500 cm-1 is ommited and peak fitting doesn't match the experimental data. Consider revising the fitting.
Answer: Thank you for your suggestion. As mentioned above, we changed the number of fitted peaks from 4 to 2. So, that problem was avoided accordingly. Please check Figure 4 in the revised manuscript.
Q7: In Section 3.3, please denote what you mean by "ultimate analysis" (Table S1, lines 143 and 160 of the main body).
Answer: Thank you for your suggestion. We have replaced “Ultimate analysis” with “Elemental analysis”. Meanwhile, we have made necessary supplementary explanations in the Materials and Methods section. Please check 3.3 Characterization of LGBCs, lines 143 and 163 in the revised manuscript and Table S1 in the Supplementary material.
Q8: Conclusions, line 460, state that "the amount of aromatic C=C bonds increases", but the reply to Q7 indicates that "neither FTIR nor Raman spectroscopy demonstrate the graphitization of the material". Please revise the fragment.
Answer: Based on your suggestion, we have made modifications to the relevant sections. Please check 4 Conclusion section in the revised manuscript.
Q9: Supplementary material, "Kf is the Freundlich affinity coefficient (mg(1-n)Ln/g)": if measurement units are denoted in the brackets, why does the notation contains dimensionless parameter "n"? What is Ln?
Answer: Thank you for your suggestion. There were errors in our previous presentation, and we missed the unit's superscript. The unit for Kf is (mg(1-n)Ln)/g, which can be confirmed from the References and formula derivation [1, 2]. We have now made the correction. Accordingly, Ln should be Ln, a part of the unit of Kf ((mg(1-n)Ln)/g). Please check the Supplementary material.
Q10: Supplementary material, section 1.3: please denote that the measurement units of Gibbs free energy, enthalpy, and entropy are Joules.
Answer: As shown in Table 6, the units for Gibbs free energy, enthalpy, and entropy are kJ/mol, kJ/mol, and J/(mol K) respectively. These values were calculated based on the data of adsorption capacity changing with ambient temperature (Eg. S5-S10). Accordingly, the values we regulated and measured were the adsorption amount and the environmental temperature, not Joules. Please check Table 6 in the revised manuscript and the Supplementary material.
References:
- Kılıç, M.; Kırbıyık, Ç.; ÇepelioÄŸullar, Ö.; Pütün, A. E., Adsorption of heavy metal ions from aqueous solutions by bio-char, a by-product of pyrolysis. Applied Surface Science 2013, 283, (0), 856-862.
- Wang, Z.; Liu, G.; Zheng, H.; Li, F.; Ngo, H. H.; Guo, W.; Liu, C.; Chen, L.; Xing, B., Investigating the mechanisms of biochar’s removal of lead from solution. Bioresource Technology 2015, 177, (0), 308-317.
